# Conditioned Media from Human Pulp Stem Cell Cultures Improve Bone Regeneration in Rat Calvarial Critical-Size Defects

**DOI:** 10.3390/jfb14080396

**Published:** 2023-07-25

**Authors:** Leonardo Fernandes Buss, Gustavo Sigrist de Martin, Elizabeth Ferreira Martinez, Isabela Amanda de Abreu Araújo Porcaro Filgueiras, José Luiz Magnabosco, Bruno Frenhan Alves, Bruno de Macedo Almeida, Tatiana Kotaka, Marcelo Lucchesi Teixeira, José Ricardo Muniz Ferreira, Daniel Navarro da Rocha, Raul Canal, Antonio Carlos Aloise, Lexie Shannon Holliday, André Antonio Pelegrine

**Affiliations:** 1Faculdade de Odontologia São Leopoldo Mandic, Campinas 13045-755, SP, Brazil; leonardobuss@hotmail.com (L.F.B.); gustavosigrist@uol.com.br (G.S.d.M.); isabela.a.filgueiras@gmail.com (I.A.d.A.A.P.F.); joseluizmagnabosco@hotmail.com (J.L.M.); bruno_frenhan@yahoo.com.br (B.F.A.); bma_natal@hotmail.com (B.d.M.A.); tatykotaka@gmail.com (T.K.); 2Division of Cell Biology, Faculdade São Leopoldo Mandic, Campinas 13045-755, SP, Brazil; elizabeth.martinez@slmandic.edu.br; 3Division of Prosthodontics, Faculdade São Leopoldo Mandic, Campinas 13045-755, SP, Brazil; marceloltx@gmail.com; 4RCrio Bioengenharia, Campinas 13098-324, SP, Brazil; josericardo@r-crio.com (J.R.M.F.); navarro@r-crio.com (D.N.d.R.); 5ANADEM, Brasília 70322-901, DF, Brazil; presidencia@anadem.org.br; 6Division of Oral Implantology, Faculdade São Leopoldo Mandic, Campinas 13045-755, SP, Brazil; aloiseac@gmail.com; 7Department of Orthodontics, University of Florida, Gainesville, FL 32611, USA; sholliday@dental.ufl.edu

**Keywords:** bone regeneration, stem cells, osteogenesis, conditioned media

## Abstract

The aim of this study was to test whether lyophilized conditioned media from human dental pulp mesenchymal stem cell cultures promote the healing of critical-size defects created in the calvaria of rats. Prior to the surgical procedure, the medium in which dental pulp stem cells were cultured was frozen and lyophilized. After general anesthesia, an 8 mm diameter bone defect was created in the calvaria of twenty-four rats. The defects were filled with the following materials: xenograft alone (G1) or xenograft associated with lyophilized conditioned medium (G2). After 14 or 42 days, the animals were euthanized, and the specimens processed for histologic and immunohistochemical analysis. Bone formation at the center of the defect was observed only in the G2 at 42 days. At both timepoints, increased staining for VEGF, a marker for angiogenesis, was observed in G2. Consistent with this, at 14 days, G2 also had a higher number of blood vessels detected by immunostaining with an anti-CD34 antibody. In conclusion, conditioned media from human dental pulp mesenchymal stem cell cultures had a positive effect on the regenerative process in rat critical-size bone defects. Both the formation of bone and enhancement of vascularization were stimulated by the conditioned media.

## 1. Introduction

Bone defects are created by different etiological factors, such as tumors, congenital deformities, osteo-metabolic diseases, infections, and trauma [1,2]. Moreover, in dentistry, tooth loss is one of the most prevalent causes of alveolar bone deficiency, since ridge remodeling promotes a reduction of bone in the horizontal and/or vertical dimensions of the jaw [3,4]. Inadequate bone quantity, caused by resorption of the residual bone after tooth extraction, may render osseointegrated implants for anchoring rehabilitation prostheses hard to install [5].

In order to enhance bone formation, autogenous grafts have been proposed for use in such defects, being considered the biological gold standard due to their osteo-inductive, osteo-conductive and osteogenenic properties [6]. However, due to the restricted bone availability of intraoral donor regions, as well as the tissue stress produced by obtaining autogenous tissue, the use of bone substitutes has increased [7,8].

Bone substitute biomaterials can be obtained from a variety of sources, and are categorized as allogeneic, xenogeneic, or alloplastic according to their origin. The Bio-Oss^®^ xenograft is a bone substitute widely used in alveolar reconstruction. This material, however, has only osteo-conductive potential and no osteogenic or osteo-inductive capacity [9]. As a result, research on bone tissue engineering, including the use of stem cells and/or osteo-inductive growth factors in combination with an osteo-conductive scaffold, has gained attention as an alternative for the repair of significant bone lesions [10]. In this context, mesenchymal stem cells from different origins are available in adult individuals and have osteogenic capacity [11,12,13,14]. In animal models of bone defects, studies have demonstrated that these cells, when associated with appropriate scaffolds, are more effective in promoting new bone than are scaffolds alone [15,16,17].

Some evidence in the scientific literature has pointed out that, in addition to using mesenchymal stem cells, the medium in which these cells are cultured (e.g., conditioned culture medium) can improve bone regeneration [18], due to the presence of growth factors as well as vesicles. Recent studies demonstrate that extracellular vesicles (EVs) are released by a variety of cells in the culture medium and may promote cell–cell and cell–matrix communication [19]. These vesicles are derived either directly from the plasma membrane (microsomes) or from fusion of multivesicular bodies with the plasma membrane (exosomes). They have a size range from 30 to 250 nm and are produced by a variety of cells under both physiological and pathological situations [20,21]. EVs are considered an important mechanism of communication between cells, helping to control the immune response [22] as well as other events, including bone remodeling [23].

Stem cells release cytokines, growth factors and chemokines, as well as EVs. These soluble factors may act directly in stimulating regeneration [24]. In this context, the culture medium where stem cells are cultured may improve osteogenesis [24] by allowing for a transient secretion of biologically active regenerative and immunomodulatory molecules, which has been verified by in vitro and in vivo studies [25,26]. Thus, the purpose of this histological and immunohistochemical study was to assess the efficacy of a lyophilized conditioned culture medium, obtained by the culture of mesenchymal stem cells (MSCs) derived from dental pulp, for bone regeneration in critical defects produced in the calvaria of rat.

## 2. Materials and Methods

### 2.1. Pulp-Derived Stem Cells

The human dental pulp stem cell lineage used in this study came from Lonza (Lonza, catalog #PT-5025, Cohasset, MA, USA). In the first 24 h, the cells were cultured in Dulbecco’s Modified Eagle Medium (DMEM, Gibco, Billings, MT, USA) containing 10% fetal bovine serum (Gibco) and 1% antibiotic-antimycotic solution (penicillin-streptomycin) (Sigma, St. Louis, MO, USA), plated in 75 cm^2^ plastic culture flasks and incubated at a density of 110 cells/mm^2^. About 24 h after plating, the cell cultures were supplemented with 50 µM ascorbic acid (Sigma, St. Louis, MO, USA), 10 mM β-Glycerophosphate (Sigma, St. Louis, MO, USA) and 0.1µM dexamethasone (Sigma, St. Louis, MO, USA). The cells were incubated under standard cell culture conditions (37 °C, 95% humidity, and 5% CO_2_) for 4 days, until the cells reached 70% of confluence.

The conditioned medium was collected and frozen at −20 °C for 24 h. Afterwards, the obtained freeze-conditioned-media were snap-frozen in liquid nitrogen and transferred to a lyophilizer (Christ, model Alpha 1-2, Osterode, Germany) and lyophilized at −55 °C under a vacuum condition of 0.040 m Bar for 48 h. Each 1 mL of fresh conditioned media yielded 10 mg of lyophilized conditioned medium in powder form.

### 2.2. Reconstitution of the Lyophilized Culture Medium and Incorporation into the Scaffold (Bio-Oss)

The freeze-dried conditioned medium in powder form was stored at −80 °C for up to 3 months. At the time of experiment, the lyophilized powder was dissolved in deionized water at a concentration of 20 mg/mL and then filtered through a 0.2 μm filter to produce a sterile final product, and then used right away.

The scaffold used in this study was the Bio-Oss bone substitute bimaterial (Bio-Oss^®^ small—Geistlich Pharma AG, Wolhusen, Switzerland). It is a xenograft hydroxyapatite of bovine origin that is commonly used clinically to fill bone defects. Sequentially, 100 mg of xenograft granules was drip-associated with the reconstituted lyophilized culture medium in 12-well cell culture plates (Corning, New York, NY, USA) immediately before the surgery procedure.

### 2.3. Animals and Study Design

Twenty-four male Wistar rats (Rattus norvegicus albinus) were used in this study, with prior approval from the Research Ethics Committee for Animal Experimentation of the Faculdade São Leopoldo Mandic (ethical protocol approval n. 2020/010). The study was carried out in compliance with the ARRIVE (Animal Research: Reporting of In Vivo Experiment) guidelines. Considering the possibility of human cell rejection, an immunosuppression protocol was started 10 days before the experiment, as described by Lekhooa et al. (2017) [27], and followed until euthanasia. The animals were kept under controlled conditions of temperature and lighting, with a 12-h light–dark cycle, balanced food, and water ad libitum.

The animals were randomly divided into two groups, according to the treatment: Group 1 (G1) and Group 2 (G2). In G1 (*n* = 12), bone defects were filled with Bio-Oss xenograft (Bio-Oss^®^ small—Geistlich Pharma AG, Wolhusen, Switzerland) and covered by collagen membrane (Bio-Gide^®^—Geistlich Pharma AG, Wolhusen, Switzerland). In G2 (*n* = 12), bone defects were filled by Bio-Oss^®^ small particles in addition to conditioned MSCs culture medium (CM-MSCs) prepared as described below and covered by Bio-Gide^®^ collagen membrane. The animals were euthanized after 14 days (T1) or 42 days (T2).

### 2.4. Surgical Protocol

Surgical procedures were performed using 10% ketamine hydrochloride (Dopalen Vetbrands, Jacareí, Brazil) and 2% xylazine hydrochloride (Rompun Bayer, São Paulo, SP, Brazil), respecting the principles of biosafety to avoid infectious processes in surgical areas.

The animals were submitted to trichotomy in the region of the calvaria and subsequent antisepsis of the area with 2% chlorhexidine solution (Biodinamica Química e Farmacêutica LTDA, Ibiporã, Brazil). A local anesthesia was administered by means of 2% Lidocaine Hydrochloride with epinephrine 1:100,000 (Alphacaine^®^ 100—DFL, Indústria e Comércio S.A., Rio de Janeiro, Brazil) to promote local ischemia. To access the cranial cap of the animal, a 15 mm long linear incision was made using a scalpel blade n˚15C (Swann-Morton, Sheffield, England) in the integument covering the skull, followed by total flap detachment using a Molt 2/4 detacher (Maximus, Industry and Commerce of Hospital and Dental Instruments LTDA, Contagem, Brazil). The flaps were kept folded with the aid of Senn Muller retractor (Rhosse, Ribeirão Preto, Brazil) used to expose the bone surface. The critical bony defect was made using an 8.0 mm diameter trephine drill (Maximus, Contagem, Brazil), crossing the entire bone thickness of the diploe. The bone fragment was removed, exposing the meninges at the bottom of the defect. Sequentially, the defect was filled with biomaterials according to the determined groups and covered with collagen membranes measuring 10 mm × 10 mm (Figure 1). The flap was repositioned and sutured with 5 simple equidistant stitches, using 5-0 needle-threated nylon (Ethicon^®^, Johnson & Johnson do Brasil Indústria e Comércio de Produtos para Saúde LTDA, São José dos Campos, Brazil).

After surgery, the animals of both groups were placed in cages and received postoperative medication, namely, dipyrone 0.5 g/mL (Algivet^®^—VETNIL, Louveira, Brazil) intraperitoneally and Rifamycin SV sodium spray 10 mg/mL (EMS, Pharma, Hortolândia, Brazil) both for 1 week. After 14 days (T1) or 42 days (T2) of the surgical procedure, the animals were euthanized by means of an overdose of the anesthetic isoflurane 2% until confirmation of respiratory arrest and loss of heart rate. To collect the skullcap, the animals were guillotined, the soft tissue was removed with scissors and the areas of interest immersed in 10% formaldehyde.

### 2.5. Histologic Processing

The calvarias were demineralized in 20% formic acid, dehydrated and embedded in histological paraffin in order to perform 4-µm-thick cuts in the central region of the defects. The samples were stained with hematoxylin-eosin, and then mounted as photomicrographs on resin slides. Images were captured on a computerized imaging system (AxioVision rel 4.8, Carl Zeiss, Oberkochen, Germany) coupled to the Axioskop 2 Plus light microscope (Carl Zeiss, Oberkochen, Germany).

### 2.6. Immunohistochemistry

From formalin-fixed paraffin-embedded tissue samples, 4 μm sections were cut and mounted on aminopropyltriethoxysilane-coated slides. The sections were deparaffinized and hydrated in decreasing concentrations of ethanol. The specimens were then incubated in a methanol solution of 3% hydrogen peroxide (Dinâmica, Diadema, SP, Brazil), to block endogenous peroxidase, for 30 min at room temperature. Antigen retrieval was achieved by boiling the slides in a steamer immersed in a citrate buffer (pH 6.0, 8 min, Sigma, St Louis, MO, USA). Subsequently, the sections were incubated with the primary antibody (CD-34, 1:50 from abcam, catalog number #ab82389, and VEGF, 1:50 from Abcam, catalog number # ab1316) at 4 °C overnight.

The secondary antibodies used were a dextran polymer-conjugated two-step (Envision^®^, Dako Carpinteria, CA, USA) and a biotinylated peroxidase conjugated streptavidin system (Advanced HRP-link^®^, Dako,), respectively, for CD34 and VEGF, for 30 min at 37 °C. Positive controls for each antibody were used as suggested by their suppliers. Specimens were processed without primary antibodies for use as negative controls.

Aminoethylcarbazole with 3,3’-diaminobenzidine tetrahydrochloride (DAB, Dako, Carpinteria, CA, USA) was used as the chromogen for 10 min at 37 °C.

The sections were counterstained with Mayer’s hematoxylin for 3 min at room temperature and visualized using a Axioskop 2 Plus light microscope (Carl Zeiss, Oberkochen, Germany).

### 2.7. Histologic, Histomorphometric and Immunohistochemical Analysis

Descriptive histological analyzes were performed considering the presence of loose connective tissue, vascularization, inflammatory infiltrate, multinucleated giant cells, newly formed bone and residual particles of biomaterials.

For bone formation at the center of the defect, the area of newly formed bone was traced using ImageJ software [28,29] (National Institutes of Health, Bethesda, MD, USA) on photomicrographs taken at 100× magnification. To allow measurement in the center of the defect, the beginning points of the native bone were located as reference points at both sides of the histological slides. All results were scored in square micrometers and then expressed as a percentage of the total area.

For the inflammatory process, a classification score was adopted considering the extent of the inflammatory process in the defect area from 0 to 3, with 0 being absent, 1 being discrete (up to 25%), 2 being moderate (25% to 50%), and 3 being intense (>50%).

For vascularization, the expressions of VEGF and CD34 were analyzed. To verify the VEGF expression, a semiquantitative approach was used to define the percentage of cell positivity, which was graded 0 to 3, where 0 corresponded to less than 10% positive cells, grade 1 ranged from 10% to 25% positive cells, grade 2 ranged from 25% to 50% positive cells, and grade 3 contained greater than 50% positive cells. For CD34 positive expression, three randomized images from each slide were captured with 400× magnification. A vessel was considered a new blood vessel when the endothelium surrounding the lumen reacted immunopositively for CD34. The vessels were counted by using the ImageJ software [28,29] (National Institutes of Health, Bethesda, MD, USA).

### 2.8. Statistical Analysis

Descriptive and exploratory analyses of all data were performed. An asymmetric distribution was observed, and the data were analyzed by means of the Mann–Whitney test. Analyses were conducted using the R software (version 4.2. 3) with a significance level of 5%.

## 3. Results

In both groups, controls (G1) or those treated with conditioned media (G2), the defects were characterized by loose immature connective tissue. Particle surfaces were surrounded by fibroblastic cells after 14 days, and this continued until day 42 (Figure 2). Bone formation at the central region of the defect of G2 was observed, specifically, at 42 days (Table 1). No bone formation was detected in the central region of the G1 defects at either time point. A slight mononuclear inflammatory infiltrate was evident in G1 and G2 after 14 days. This had cleared by day 42. Multinuclear giant cells were not detected in either group at either time point.

VEGF was detected within both G1 and G2 calvaria at both 14 and 42 days by immunohistochemistry (Figure 3). The expressions are depicted in Figure 3 and Table 2. VEGF was present in all connective tissue. Staining was more intense in G2 (Figure 3). In addition, a significantly higher score of cells stained positive in G2 compared with G1 at both day 14 and day 42 (Table 2). After 14 days, the score of cells positive for VEGF staining in G2 was dramatically higher than in G1. After 42 days, the score of VEGF positive cells significantly decreased, but remained significantly higher than in G1.

As a marker for new blood vessel formation, we stained the calvaria with an anti-CD34 antibody. The results of CD34 immunostaining are presented in Figure 4 and Table 3. After 14 days, a higher number of new blood vessels was observed in G2 when compared to G1 (*p* = 0.0216). No difference was observed in blood vessel count after 42 days for each analyzed group (*p* = 0.5309). For G1, an increase in new blood vessel number was identified after 42 days, as compared to G2 (*p* = 0.0472).

## 4. Discussion

In this study, we present evidence for the first time that media conditioned by stem cells obtained from dental pulp stimulate osteogenesis in a rat calvarial critical-size defect model. This suggests that soluble factors released by these stem cells might prove useful for osteogenic therapy.

The findings demonstrated that conditioned media (G2) stimulated bone growth at the central region of the critical-size defect after 42 days. Furthermore, when compared to defects exclusively treated with hydroxyapatite biomaterial (G1), after 14 days of postoperative duration, G2 had a greater expression of VEGF, as well as CD34. This might be attributed to the stem-cell-derived EVs in lyophilized conditioned media, which contain factors that regulate cell mobilization and osteogenic differentiation as well as angiogenesis [24,30]. In the case of G1, when only the hydroxyapatite-containing Bio-Oss biomaterial was used, no bone growth was verified in the center of the defect. The absence of osteo-inductive elements in the inorganic Bio-Oss biomaterial, together with the complexity of the defect (i.e., a critical-size defect), might have contributed to these findings.

We show that VEGF expression was higher in G2 than in G1 at both time points. VEGF is an important stimulator of angiogenesis and bone formation [31]. Consistent with this, staining of new blood vessels with CD34 was higher at 14 days in G2. This indicates that the conditioned media has elements that stimulate both mineralized bone formation and angiogenesis.

Adult stem cell treatment has been extensively researched for use in tissue engineering. Bone marrow mesenchymal stem cells (BM-MSC) are capable of osteogenic development [32]. Even the use of fresh bone marrow (i.e., without extensive manipulation) can lead to an improvement in bone regeneration [33]. The approach of collecting, expanding, and differentiating BM-MSC is a well-established procedure with predictable results. However, there are certain clear barriers to using bone marrow, including donor site morbidity and the risk of contamination. In this regard, mesenchymal stem cells produced from various sources, such as the pulp of permanent or deciduous teeth, have emerged as an alternative for tissue engineering [13,34], owing to the simplicity with which the cells can be obtained when compared to bone marrow. Similarly to marrow stem cells, stem cells from dental pulp have been shown to display osteogenic differentiation [35].

Initially, stem cell therapy was conceived as an approach in which a multipotential stem cell would be delivered to a site of injury or pathology, where it would differentiate to serve directly in the repair or healing process [36]. However, it has often been found that, when stem cells contributed to the healing process, they were not present at the site of healing. This suggested that they were releasing a soluble factor or factors that were beneficial [36]. In recent years, numerous studies have appeared showing that soluble factors produced by stem cells, when introduced into an experimental system, produce benefits [37,38,39]. This study is another example of this general finding: that soluble factors from stem cells are therapeutic.

Stem cells secrete a variety of proteins, peptides, RNA and lipid mediators that can be concentrated, frozen, or even lyophilized without losing activity [40], allowing them to be used as biological and immunomodulator active agents [26]. Since stem cell cultures and their derivatives, including culture media, can be used in regenerative procedures, this study explored the effect of lyophilized conditioned media derived from dental pulp mesenchymal stem cell culture, in combination with Bio-Oss, on critical-size defects in rat calvaria.

Extracellular vesicles (EVs) are small membrane vesicles that have recently been identified as essential mediators of cell-to-cell communication pathways, mediating the complete immune response in both pathological and healthy contexts [41]. EVs have also been identified as important regulators of bone remodeling [23]. As they can be found in cell culture supernatants and in different biological fluids, the use of culture media, which delivers nutrients and proteins to the cells while also containing the EVs produced by the cells, may be an alternative for use in tissue regeneration [42]. In fact, Takeuchi et al. [28] revealed that bone marrow stem-cell-derived EVs, delivered in culture media, can enhance angiogenesis and bone regeneration.

The establishment of critical-size bone defects in animal models is crucial in preclinical studies for evaluating the osteogenic potential of therapies aimed at improving bone healing, because it does not result in considerable spontaneous regeneration. Thus, by using critical-size defect experimental models, the real contribution of the applied therapies can be observed. According to Song et al. [43], a bone defect of 8 mm in the center of the rat calvaria is considered critical. In this regard, defects with smaller diameters, those up to 6 mm, are non-critical since they result in a large amount of spontaneous bone growth, particularly after 8 weeks [44].

A number of studies investigated the repair of bone defects in the femur or tibia of rats; however, these bones are subjected to mechanical pressure, which can influence the results. This variable is eliminated with the use of calvarial bone defects. As a result, the 8 mm defect model created in the calvaria of rats was used in this study. The use of osteo-conductive biomaterials alone may not result in effective bone regeneration of bone lesions of this size. In this context, the scientific literature has reported the use of stem cells and/or growth factors as bone healing potentiators, typically in conjunction with osteo-conductive biomaterials [45,46,47]. However, it is important to note that other biomaterials, such as allografts, which are associated even with better outcomes for alveolar ridge preservation [48], should be tested for cell therapy/tissue engineering purposes, even though a xenograft was chosen as the scaffold for the current study.

It is important to highlight that these inductive potentials were acquired by employing lyophilized conditioned media in the surgical areas of the defect. A freeze-drying method for lyophilization of conditioned culture media was used in the present study, one similar to the protocol adopted by Peng et al. [49]. The conditioned media has traditionally been used in a liquid state, that is, for immediate use in cell culture supplementation. The use of cell-free delivery therapy when combined with mineralized biomaterial scaffolds, such as those used in this study, offers numerous benefits, including safety, ease of control and manipulation, simple storage, economic and practical advantages, mass production and customization based on the desired effects for clinical applications [50]. Therefore, the study of lyophilized cell-free conditioned media seems to be of major importance and a significant trend. The present study was designed to test whether the conditioned media would contribute to bone regeneration. As the results showed, the null hypothesis was rejected (i.e., the conditioned media promoted bone regeneration). Future investigations are needed to examine the mechanisms by which the conditioned media influenced bone regeneration.

As a future perspective, studies evaluating possible benefits of the combined use of conditioned medium together with new titanium implants and surface modifications, such as nano-surfaces obtained by oxidation methods [51,52], are needed. The combined use of conditioned media with modified titanium implants might further improve levels of osseointegration.

## 5. Conclusions

The results of the present study demonstrated that lyophilized conditioned culture media derived from mesenchymal stem cells increased vascularization and had beneficial effects on bone regeneration favorable for use in tissue engineering. As a result, functionalizing a hydroxyapatite scaffold with mesenchymal stem-cell-conditioned media is a realistic, effective and potentially therapeutically relevant method for enhancing bone repair. The presented results should encourage future investigations in the field of bone regeneration and osseointegration that might generate new products and therapeutic protocols.

## Figures and Tables

**Figure 1 jfb-14-00396-f001:**
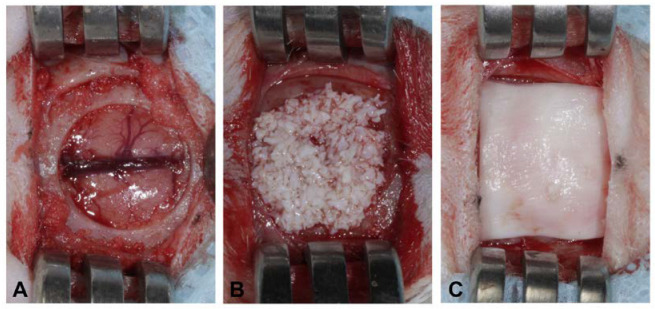
Procedure for critical-size defect study. (**A**) Rat calvarial critical-size defect. (**B**) Critical-size defect filled with Bio-Oss. (**C**) Bio-Oss defect covered with collagen membrane.

**Figure 2 jfb-14-00396-f002:**
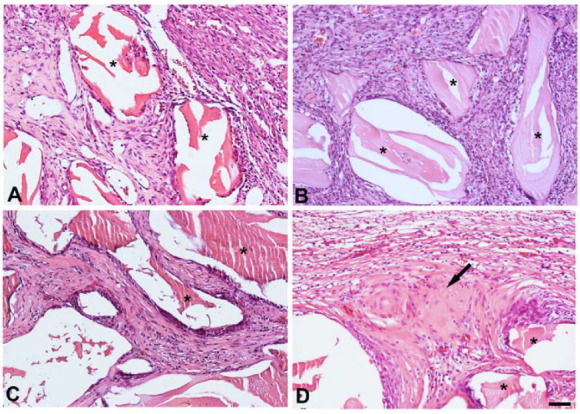
Significant bone formation in critical-size defects is triggered by conditioned media after 42 days. Photomicrographs of histological sections showing remaining hydroxyapatite biomaterial (*) and newly formed bone (arrow). Legend: (**A**) G1, 14 days; (**B**) G2, 14 days; (**C**) G1, 42 days; (**D**) G2, 42 days. Scale bar = 50 μm.

**Figure 3 jfb-14-00396-f003:**
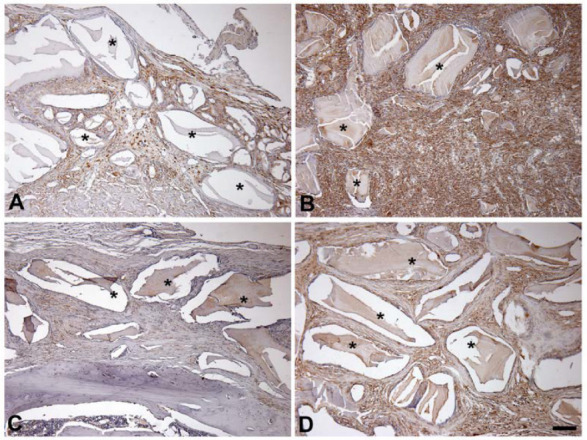
VEGF expression in calvarial critical-size bone defects is higher at 14 and 42 days postoperative: (**A**) G1, 14 days; (**B**) G2, 14 days; (**C**) G1, 42 days; (**D**) G2, 42 days. Photomicrographs of histological sections showing remaining hydroxyapatite biomaterial (*) and VEGF positive cells (dark brown). Scale bar = 50 μm.

**Figure 4 jfb-14-00396-f004:**
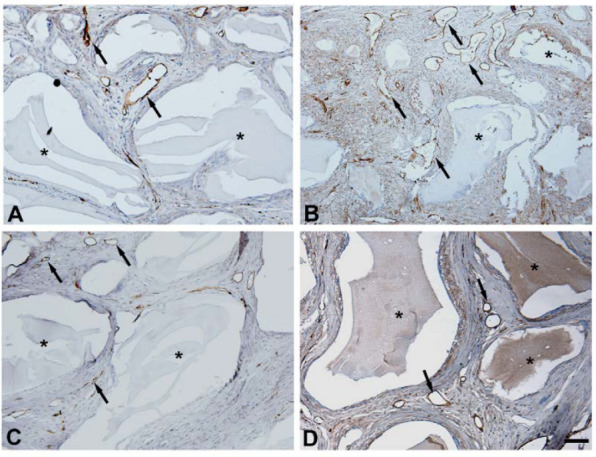
Analysis of CD34 expression in different groups at 14 and 42 days postoperative: (**A**) G1, 14 days; (**B**) G2, 14 days; (**C**) G1, 42 days; (**D**) G2, 42 days. Photomicrographs of histological sections showing remaining hydroxyapatite biomaterial (*) and new blood vessels (arrows). Scale bar = 50 μm.

**Table 1 jfb-14-00396-t001:** New bone formation percentage (%) at 14 and 42 days postoperative.

Group	Days			
	14		42	
	Median (Interquartile Interval)	Minimum–Maximum	Median (Interquartile Interval)	Minimum–Maximum
G1	0 (0)	0/0	0 (0)	0/0
G2	0 (0)	0/0	8.44 (1.47)	6.97/17.21

**Table 2 jfb-14-00396-t002:** VEGF scores in both groups at 14 and 42 days postoperative.

Group	Days				*p*-Value
	14		42		
	Median (Interquartile Interval)	Minimum–Maximum	Median (Interquartile Interval)	Minimum–Maximum	
G1	1 (0.2) Ab	1–2	1 (0) Ab	1–1	0.5637
G2	3 (0) Aa	3–3	2 (0) Ba	2–2	0.0209
*p*-valor	0.0209		0.0209		

Legend: Scores: 0 = absent, 1 = up to 25% (mild), 2 = 25% to 50% (moderate), and 3 = more than 50% (intense) of VEGF expression. Distinct letters (upper case horizontally and lower case vertically) indicate statistically significant differences (*p* < 0.05).

**Table 3 jfb-14-00396-t003:** New blood vessels (detected by anti-CD34 staining) counted in different groups at 14 and 42 days postoperative.

Group	Days				*p*-Value
	14		42		
	Median (Interquartile Interval)	Minimum–Maximum	Median (Interquartile Interval)	Minimum–Maximum	
G1	25 (8) Bb	23–34	37 (7) Aa	30–48	0.0472
G2	43 (9) Aa	32–48	38 (6) Aa	33–64	0.7540
*p*-value	0.0216		0.5309		

Legend: Distinct letters (upper case horizontally and lower case vertically) indicate statistically significant differences (*p* < 0.05).

## Data Availability

The data presented in this study are available on request from the corresponding author.

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
