# Peer review of "Conditioned Media from Human Pulp Stem Cell Cultures Improve Bone Regeneration in Rat Calvarial Critical-Size Defects"

_jfb, 2023, doi:10.3390/jfb14080396_

Round 1

Reviewer 1 Report

The authors present novel results of lyophilized powder from stem cell extracts which are found to significantly improve healing in damaged bones of rats. The work is novel and highly important and interesting for regenerative medicine. I suggest acceptance after following points are improved in a minor revision.

1. It is not clear why in vitro studies has been skipped. Please provide better explanations for it in the manuscript.

2. The introduction is leaking about improved bone regeneration by surface-modified implants. A suitable paragraph should be added and the advantages and disadvantages in comparison to the here presented approach be discussed. 1  

3. The used chemicals should be mentioned correctly: (purity/molecular weight for polymers, manufacturer, city of manufacturing, country of manufacturing)

4. The utilized devices and software are not always mentioned correctly: (type, company, city, country)

5.  “4 μmm” on page 4 line 166, please revise it.

6.  “4oC” on page 4 line 176, please revise it.

7.     “4oC” on page 4 line 182, please revise all degree signs correctly throughout the whole paper.

8.     Space signs are missing throughout the whole manuscript before and after an equal sign.

9.     Figures 1, 2, 3, 4: Scale bars are missing. Please add them.

10.  The authors should cite the imageJ software, as stated in the requirements of imageJ.2,3

11.  Software utilized for statistical analysis is not stated.

12.  Page 9 line 301: reference missing.

13. The work of the authors is important and very important also for implant anchoring processes, which are needed to improve patient benefit and healing. Novel implants with osteoconductive properties like these 3 could benefit from this work a lot.

14.  The conclusion is far too small. It should deal with the intension why the study was conducted, the important results should be mentioned and an outlook should be given how the results can be used in the future by the researcher themselves or by others.

15. The ethical research statement for the in vivo studies as the “Institutional Review Board Statement” is missing.

References to cite:

(1)      Kozelskaya, A. I.; Rutkowski, S.; Frueh, J.; Gogolev, A. S.; Chistyakov, S. G.; Gnedenkov, S. V; Sinebryukhov, S. L.; Frueh, A.; Egorkin, V. S.; Choynzonov, E. L.; Buldakov, M.; Kulbakin, D. E.; Bolbasov, E. N.; Gryaznov, A. P.; Verzunova, K. N.; Apostolova, M. D.; Tverdokhlebov, S. I. Surface Modification of Additively Fabricated Titanium-Based Implants by Means of Bioactive Micro-Arc Oxidation Coatings for Bone Replacement. J. Funct. Biomater. 2022, 13 (4), 285. https://doi.org/https://doi.org/10.3390/jfb13040285.

(2)      Schneider, C. A.; Rasband, W. S.; Eliceiri, K. W. NIH Image to ImageJ: 25 Years of Image Analysis. Nat. Methods 2012, 9 (7), 671–675. https://doi.org/10.1038/nmeth.2089.

(3)      Abràmoff, M. D.; Magalhães, P. J.; Ram, S. J. Image Processing with ImageJ Part II. Biophotonics Int. 2005, 11 (7), 36–43.

Author Response

The authors present novel results of lyophilized powder from stem cell extracts which are found to significantly improve healing in damaged bones of rats. The work is novel and highly important and interesting for regenerative medicine. I suggest acceptance after following points are improved in a minor revision.

Answer: Thank you for your careful reading and suggestions. Please refer to the revised version of the manuscript, where the new information is highlighted in yellow.

  1. It is not clear why in vitro studies has been skipped. Please provide better explanations for it in the manuscript.

Answer: In vitro studies corroborating the secretion of biologically active regenerative and immunomodulatory molecules in the culture medium are cited in line 78.

  1. The introduction is leaking about improved bone regeneration by surface-modified implants. A suitable paragraph should be added and the advantages and disadvantages in comparison to the here presented approach be discussed. 1

Answer: The reason why we haven’t inserted a bone formation level comparison between titanium implants surface modifications and the conditioned medium, in the Introduction section, is related to the fact that, in the present study, the medium was associated with a bone substitute biomaterial instead of a metallic implant. However, now we inserted a discussion about the possible benefits of the combined use of conditioned medium together with titanium implants surface modifications, for osseointegration purposes, at the Discussion section (lines 361-364). We also used the reference 1 suggested by the referee at this point.

  1. The used chemicals should be mentioned correctly: (purity/molecular weight for polymers, manufacturer, city of manufacturing, country of manufacturing)

Answer: This information was inserted.

  1. The utilized devices and software are not always mentioned correctly: (type, company, city, country)

Answer: This information was inserted.

  1. “4 μmm” on page 4 line 166, please revise it.

Answer: It was corrected.

  1. “4oC” on page 4 line 176, please revise it.

Answer: It was corrected.

  1. “4oC” on page 4 line 182, please revise all degree signs correctly throughout the whole paper.

Answer: It was corrected (in the whole text).

  1. Space signs are missing throughout the whole manuscript before and after an equal sign.

Answer: It was corrected.

  1. Figures 1, 2, 3, 4: Scale bars are missing. Please add them.

Answer: It was done on Figures 2, 3 and 4.

  1. The authors should cite the imageJ software, as stated in the requirements of imageJ.2,3

Answer: It was done

  1. Software utilized for statistical analysis is not stated.

Answer: It was done

  1. Page 9 line 301: reference missing.

Answer: It was inserted

  1. The work of the authors is important and very important also for implant anchoring processes, which are needed to improve patient benefit and healing. Novel implants with osteoconductive properties like these 3could benefit from this work a lot.

Answer: We inserted the discussion about the possible benefits of the combined used of conditioned medium together with novel implants surface modifications, for osseointegration purposes, at the Discussion section (lines 361-364).

  1. The conclusion is far too small. It should deal with the intension why the study was conducted, the important results should be mentioned and an outlook should be given how the results can be used in the future by the researcher themselves or by others. 

Answer: One more phrase was added at the end of the Conclusion.

  1. The ethical research statement for the in vivostudies as the “Institutional Review Board Statement” is missing. 

Answer: It was highlighted in the “Animals and Study Design” section. Besides that, I can share the signed document (if necessary). Moreover, as it is in Portuguese, I can also translate it to English (if necessary).

References to cite:

(1)      Kozelskaya, A. I.; Rutkowski, S.; Frueh, J.; Gogolev, A. S.; Chistyakov, S. G.; Gnedenkov, S. V; Sinebryukhov, S. L.; Frueh, A.; Egorkin, V. S.; Choynzonov, E. L.; Buldakov, M.; Kulbakin, D. E.; Bolbasov, E. N.; Gryaznov, A. P.; Verzunova, K. N.; Apostolova, M. D.; Tverdokhlebov, S. I. Surface Modification of Additively Fabricated Titanium-Based Implants by Means of Bioactive Micro-Arc Oxidation Coatings for Bone Replacement. J. Funct. Biomater. 202213 (4), 285. https://doi.org/https://doi.org/10.3390/jfb13040285.

(2)      Schneider, C. A.; Rasband, W. S.; Eliceiri, K. W. NIH Image to ImageJ: 25 Years of Image Analysis. Nat. Methods 20129 (7), 671–675. https://doi.org/10.1038/nmeth.2089.

(3)      Abràmoff, M. D.; Magalhães, P. J.; Ram, S. J. Image Processing with ImageJ Part II. Biophotonics Int. 200511 (7), 36–43.

Answer: These 3 references were used.

- Please, see the new version of the manuscript attached (all modifications are highlighted).

Reviewer 2 Report

To authors:

The manuscript is scientifically very well structured and executed for which I would like to congratulate the authors.

The manuscript present important and interesting results in beneficial effects on bone regeneration for use in tissue engineering.

I congratulate and encourage the authors to deepen the scientific theme described.

Author Response

Thank you for your review and feedback.

Reviewer 3 Report

Nice article. This Reviewer suggests the following to improve the paper:

You state that BioOss is the most widely used Xenograft, but provide no cite to prove your statement (cite No. 9 does not prove your statement).

You have no mention of the use of allografts of use in sockets, a discussion of this and a comparison to xenografts should be included in the Discussion.

You have no arrows in Fig.4.

Author Response

Thank you for your review and suggestions. Please refer to the revised version of the manuscript, where the new information is highlighted in yellow. We agree that we haven’t provide citations to prove the statement that Bio-Oss is the most widely used xenograft. So, we removed the word (“most”).

Concerning the mention about the use of allografts in sockets (and a comparison with xenografts), we inserted some information at the Discussion section (lines 338-341).

Regarding Figure 4, arrows were placed.

- Please, see the new version of the manuscript attached (all modifications are highlighted).

Reviewer 4 Report

The work deserves to be accepted after minor grammatical or expression corrections. It offers good alternatives for bone tissue regeneration.

Line 84. This sentence is missing the verb. The human dental pulp stem cell lineage (Lonza, USA, catalog #PT-5025).

 From line 105. It is not presented in the first place to BIO-Oss, nor is its origin stated, nor is it how it was prepared.

 Line 56. It is not clearly indicated after surgery, the treatment time, if it was until they were sacrificed, in each of the groups, or to both groups for the same period of time.

 Line 173. The immunohistochemical procedure is not clearly described, if the samples required a different preparation prior to embedding them in paraffin, and if after cutting the sections, the sections were not immobilized in some special type of glass. The authors must report the entire procedure so that any researcher can repeat it.

no comments

Author Response

The work deserves to be accepted after minor grammatical or expression corrections. It offers good alternatives for bone tissue regeneration.

Answer: Thank you for your careful reading and suggestions. Please refer to the revised version of the manuscript, where the new information is highlighted in yellow.

Line 84. This sentence is missing the verb. The human dental pulp stem cell lineage (Lonza, USA, catalog #PT-5025).

Answer: The sentence was corrected.

 From line 105. It is not presented in the first place to BIO-Oss, nor is its origin stated, nor is it how it was prepared.

Answer: This description was inserted (lines 106-108).

 Line 56. It is not clearly indicated after surgery, the treatment time, if it was until they were sacrificed, in each of the groups, or to both groups for the same period of time.

Answer: This information was inserted (lines 158-160).

 Line 173. The immunohistochemical procedure is not clearly described, if the samples required a different preparation prior to embedding them in paraffin, and if after cutting the sections, the sections were not immobilized in some special type of glass. The authors must report the entire procedure so that any researcher can repeat it.

Answer: This information was inserted in this section.

- Please, see the reviewed manuscript attached.

Round 2

Reviewer 1 Report

The most point of the revision has been done by the authors according to the comments. I suggest the editor to accept this manuscript.